# REDUCING THE REPRESENTATION ERROR OF GAN IMAGE PRIORS USING THE DEEP DECODER

## ABSTRACT

Generative models, such as GANs, learn an explicit low-dimensional representation of a particular class of images, and so they may be used as natural image priors for solving inverse problems such as image restoration and compressive sensing. GAN priors have demonstrated impressive performance on these tasks, but they can exhibit substantial representation error for both in-distribution and out-of-distribution images, because of the mismatch between the learned, approximate image distribution and the data generating distribution. In this paper, we demonstrate a method for reducing the representation error of GAN priors by modeling images as the linear combination of a GAN prior with a Deep Decoder. The deep decoder is an underparameterized and most importantly unlearned natural signal model similar to the Deep Image Prior. No knowledge of the specific inverse problem is needed in the training of the GAN underlying our method. For compressive sensing and image superresolution, our hybrid model exhibits consistently higher PSNRs than both the GAN priors and Deep Decoder separately, both on in-distribution and out-of-distribution images. This model provides a method for extensibly and cheaply leveraging both the benefits of learned and unlearned image recovery priors in inverse problems.

## 1 INTRODUCTION

Generative Adversarial Networks (GANs) show promise as priors for solving imaging inverse problems such as inpainting, compressive sensing, super-resolution, and others. For example, they have been shown to perform as well as common sparsity based priors on compressed sensing tasks using 5-10x fewer measurements, and also perform well in nonlinear blind image deblurring (Bora et al., 2017; Asim et al., 2018). The typical inverse problem in imaging is to reconstruct an image given incomplete or corrupted measurements of that image. Since there may be many potential reconstructions that are consistent with the measurements, this task requires a prior assumption about the structure of the true image. A traditional prior assumption is that the image has a sparse representation in some basis. Provided the image is a member of a known class for which many examples are available, a GAN can be trained to approximate the distribution of images in the desired class. The generator of the GAN can then be used as a prior, by finding the point in the range of the generator that is most consistent with the provided measurements.

We use the term "GAN prior" to refer to generative convolutional neural networks which learn a mapping from a low dimensional latent code space to the image space, for example with the DCGAN, GLO, or VAE architectures (Radford et al., 2015; Bojanowski et al., 2017; Kingma & Welling, 2013). Challenges in the training of GANs involve selecting hyperparameters, like the dimensionality of the model manifold; difficulties in training, such as mode collapse; and the fact than GANs are not directly optimizing likelihood. Because of this, their performance as image priors is severely limited by representation error (Bora et al., 2017). This effect is exaggerated when reconstructing images which are out of the training distribution, in which case the GAN prior typically fails completely to give a sensible solution to the inverse problem.

In contrast, untrained deep neural networks also show promise in solving imaging inverse problems, by leveraging architectural bias of a convolutional network as a structural prior instead of a learned representation (Ulyanov et al., 2018; Heckel & Hand, 2018). These methods are independent of any training data or image distribution, and therefore are robust to shifts in data distribution that

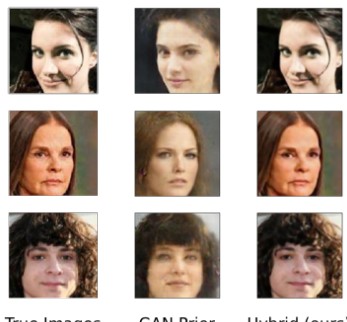

True Images     GAN Prior     Hybrid (ours)

Figure 1: Our hybrid model, a combination of a GAN prior and a Deep Decoder, has significantly less representation error than the GAN Prior alone.

are problematic for GAN priors. Recent work by Heckel & Hand (2018) presents an untrained decoder-style network architecture, the Deep Decoder, that is an efficient image representation and as a consequence works well as an image prior. In particular, it can represent images more efficiently than with wavelet thresholding. When used for denoising tasks, it outperforms BM3D, considered the state-of-the-art among untrained denoising methods. The Deep Decoder is similar to the Deep Image Prior, but it can be underparameterized, having fewer optimizable parameters than the image dimensionality, and consequently does not need any algorithmic regularization, such as early stopping.

In this paper, we propose a simple method to reduce the representation error of a generative prior by studying image models which are linear combinations of a trained GAN with an untrained Deep Decoder. We build a method that capitalizes on the strengths of both methods: we want strong performance for all natural images and not just those close to a training distribution, and we want improved performance when given images are near a provided training distribution. The former comes from the Deep Decoder, and the latter comes from the GAN. We demonstrate the performance of this method on compressive sensing tasks using both in-distribution and out-of-distribution images. For in-distribution images, we find that the hybrid model consistently yields higher PSNRs than various GAN priors across a wide variety of undersampling ratios (sometimes by 10+ dB), while also consistently outperforming a standalone Deep Decoder (by around 1 dB). Performance improvements over the GAN prior also hold in the case of imaging on far out-of-distribution images, where the hybrid model and Deep Decoder model have comparable performance. A major challenge of the field is to build algorithms for solving inverse problems that are at least as good as both learned and recently discovered unlearned methods. Any new method should be at least as good as either approach separately. The literature contains multiple answers to this question, including invertible neural networks, optimizing over all weights of a trained GAN in an image-adaptive way, and more. This paper provides a significantly simpler method to get the benefits of both learned and unlearned methods, surprisingly by simply taking the linear combination of both models.

## 2   RELATED WORK

Another approach to reducing the reconstruction error of generative models has been to study invertible generative neural networks. These are networks that are fully invertible maps between latent space an image space by architectural design. The allow for direction calculation and optimization of the likelihood of any image, in particular because all images are in the range of such networks. Consequently, they have zero representation error. While such methods have demonstrated strong empirical performance (Asim et al., 2019), invertible networks are very computationally expensive, as this recent paper used 15 GPU minutes to recover a single $64 \times 64$ color image. Much of their benefit may be obtainable by simpler and cheaper learned models.

Alternatively, representation error of GANs may be reduced through an image adaptive process, akin to using the GAN as a warm start to a Deep Image Prior. We will make comparisons to one implementation of this idea, IAGAN, in Section 4.1.1. The IAGAN method uses an entire GAN as an image model, tuning its parameters to fit a single image. This method will have negligible

representation error, and our model achieves comparable performance in low measurement regimes while using a drastically fewer optimizable parameters.

Another approach to reducing the GAN representation error could be to create better GANs. Much progress has been made on this front. Recent theoretical advances in the understanding and design of optimization techniques for GAN priors are driving a new generation of GANs which are stable during training under a wide range of hyperparameters, and which generate highly realistic images. Examples include the Wasserstein GAN, Energy Based GANs, and Boundary Equilibrium GAN (Arjovsky et al., 2017; Zhao et al., 2016; Berthelot et al., 2017). Other architectures have been proposed which factorize the problem of image generation across multiple spatial scales. For example, Style-GAN introduces multiscale latent "style" vectors, and the Progressive Growth of GANs method explicitly separates training into phases, across which the scale of image generation is increased gradually (Karras et al., 2018; 2017). In any of these examples, the demonstration of GAN quality is typically the visual appearance of the result. Visually appealing GAN outputs may still belong to GANs with significant representation errors for particular images desired to be recovered by solving an inverse problem.

## 3 METHOD

We assume that one observes a set of linear measurements $y \in \mathbb{R}^n$ of a true image $x \in \mathbb{R}^n$, possibly with additive noise $\eta$:

$$y = Ax + \eta,$$

where $A \in \mathbb{R}^{m \times n}$ is a known measurement matrix. We introduce an image model of the form $H(\vartheta)$, where $\vartheta$ are the parameters of the image representation under $H$. The empirical risk formulation of this inverse problem is given by

$$\min_{\vartheta} \|AH(\vartheta) - y\|_2^2 \qquad (1)$$

In this formulation, one must find an image in the range of the model $H$ that is most consistent with the given measurements by searching over parameters $\vartheta$. For example, Bora et al. (2017) propose to use a generative image model such as a DCGAN, GLO, or VAE, for which $\vartheta$ is a low dimensional latent code. One could also choose $\vartheta$ to be the coefficients of a wavelet decomposition, or the weights of a neural network tuned to output a single image.

In our model, we represent images as the linear combination of the output of a pretrained GAN $G_\phi(z)$ and a Deep Decoder $DD(\theta)$:

$$H(z, \theta, \alpha, \beta) = \alpha G_\phi(z) + \beta \text{DD}(\theta), \quad \vartheta = \{z, \theta, \alpha, \beta\}$$

Here, the $\phi$ are the learned weights of the GAN, which are fixed. The variables that are optimized are: $z$, the GAN latent code; $\theta$, the image-specific weights of the Deep Decoder; and scalars $\alpha$ and $\beta$.

The first part of our image model is a GAN. We demonstrate our model's performance using a BEGAN, and demonstrate the same results generalize to the DCGAN architecture. Our BEGAN has 64-dimensional latent codes sampled uniformly from $[-1, 1]^{64}$, and we choose a diversity ratio of 0.5. Our DCGAN has 100-dimensional latent codes, sampled from $\mathcal{N}(0, 0.1^2 I)$. The BEGAN prior is trained to output $128 \times 128$ pixel color images and the DCGAN prior is trained to output $64 \times 64$ color images of celebrity faces taken from the CelebA training set (Liu et al., 2015). The GANs are initially pretrained, and the only parameters that are optimized during inversion is the latent code.

The second part of our image model is a Deep Decoder. The Deep Decoder is a convolutional neural network consisting only of the following architectural elements: 1x1 convolutions, relu activations, fixed bilinear upsampling, and channelwise normalization. A final layer uses pixelwise linear combinations to create a 3 channel output. In all experiemnts, we consider a 4 layer deep decoder with $k$ channels in each layer, where $k$ is chosen so that $|\theta| < m$. Thus, our deep decoder, and even our entire model, is underparameterized with respect to the image dimensionality. The deep decoder is unlearned in that it sees no training data. Its parameters $\theta$ are estimated only at test time.

In our experiments, we found it beneficial to first partially solve (1) with $G(z)$ only and separately with $DD(\theta)$ only to find approximate minimizers $z^*$, $\theta^*$ which are then used to initialize $H$. To maintain a fair comparison between $H$ and other image models, we hold the number of *global* inversion iterations $N$ constant. We use $n_{\text{pre}} = 500$ separate inversion iterations to find $z^*$, $\theta^*$, and

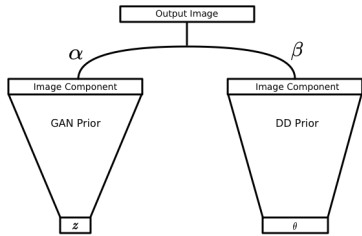

Figure 2: The model includes parameters $\alpha$, $\beta$, $\theta$, and $z$, which together comprise the image representation enforced by our Hybrid model. The final output information is a learned linear combination of the two component images.

then initialize $\alpha = 0.5$, $\beta = 0.5$, and continue with $n = 5000$ inversion iterations to optimize the parameters of $H$. To solve (1) with a GAN prior as in (Bora et al., 2017), or with a proper Deep Decoder as an image prior, we simply run $N = 5500$ inversion steps with no interruptions. We provide details on the hyperparameters used in our experiments in Section 6.1 of the Supplemental Materials.

---

**Algorithm 1** Inversion Algorithm.

---

**Require:** $n_{\text{pre}}$, the number of separate preinversion steps for $G_\phi(z)$ and $\text{DD}(\theta)$. $n$, the number of remaining inversion steps for the hybrid model. $z$ and $\theta$, random initialization parameters for $G_\phi(z)$ and $\text{DD}(\theta)$.
 1: **for** $k = 0, ..., n_{\text{pre}}$ **do**
 2:     $L_G \leftarrow \|AG_\phi(z) - y\|_2^2$
 3:     $z \leftarrow AdamUpdate(z, \nabla_z L_G)$
 4:     $L_{DD} \leftarrow \|A(\text{DD}(\theta)) - y\|_2^2$
 5:     $\theta \leftarrow AdamUpdate(\theta, \nabla_\theta L_{DD})$
 6: **end for**
 7: $\alpha \leftarrow 0.5$, $\beta \leftarrow 0.5$
 8: **for** $t = 0, ..., n$ **do**
 9:     $H \leftarrow \alpha G_\phi(z) + \beta DD(\theta)$
10:     $L \leftarrow \|AH - y\|_2^2$
11:     $z, \theta, \alpha, \beta \leftarrow AdamUpdate(\vartheta_H, \nabla_{\vartheta_H} L)$
12: **end for**

---

## 4 EXPERIMENTS

In this section, we show that our hybrid prior consistently outperforms both the individual priors, namely the GAN prior and the Deep Decoder, both for compressive sensing as well as for super-resolution.

### 4.1 COMPRESSED SENSING

In compressed sensing, one must reconstruct a signal given $m < n$ linear measurements of the image. We study compressed sensing with Gaussian measurement matrices, so that the measurement matrix $A \in \mathbb{R}^{m \times n}$ has i.i.d. Gaussian entries drawn from $\mathcal{N}(0, 1/m)$. We choose $\eta \in \mathbb{R}^m$ to be a Gaussian noise vector, normalized so that $\sqrt{\mathbb{E}[\|\eta\|^2]} = 0.1$. Our reported PSNR values are averaged over 12 random CelebA test set images.

In Figure 3, we compare the performance of a Deep Decoder, a GAN prior, and the proposed hybrid model. For our GAN prior, we use the BEGAN architecture, and we demonstrate similar results with the DCGAN architecture in the supplemental materials (Radford et al., 2015; Berthelot et al., 2017). Replicating the results in Bora et al. (2017), we observe the reconstruction quality of the GAN prior quickly plateaus as $m$ increases, illustrating its performance is limited by representation error. The Deep Decoder yields significantly higher PSNRs (sometimes by 10+ dB) when $m$ is not small. We

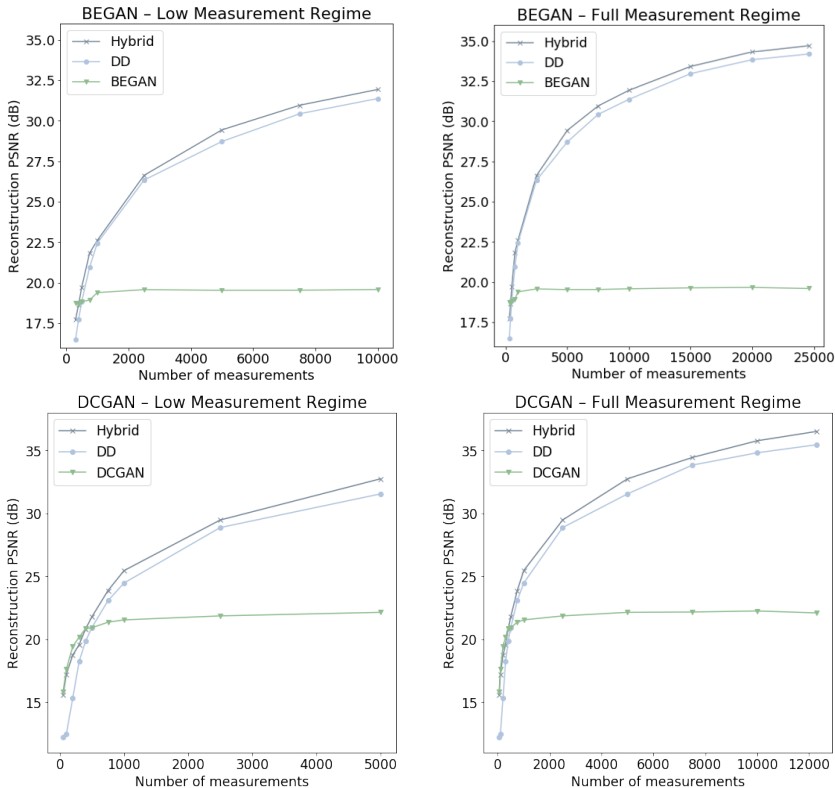

Figure 3: Reconstruction PSNRs versus measurement numbers for in-distribution test images for our hybrid model, a Deep Decoder, and a GAN prior. The left panels zoom in to the low measurement regime of the right panels. Our hybrid model is able to yield higher PSNRs than both of its components, the Deep Decoder and a GAN, on in-distribution test images, in all but the lowest measurement regime. The effect is replicated both for the BEGAN (top row) and the DCGAN (bottom row).

observe that the hybrid model yields higher PSNRs than the Deep Decoder, usually by around 1 dB, at all subsampling ratios. Further, in all but the smallest $m$ regime, the hybrid model outperforms the GAN models. is able to improve consistently when it has access to more measurements. The hybrid model is the best of all, indicating that despite the limited performance of GAN priors, they are still able to provide useful structure which the Deep Decoder cannot replicate. In particular, the Deep Decoder is biased to low frequency information in the image signal, and has a smoothing effect which is increasingly prominent with fewer measurements. Because a GAN prior learns an approximation of the true data manifold, it is biased to represent semantically relevant features of variation in the images in its training dataset. In the case of celebrity faces, the GAN prior can well represent features like eyes, noses, and mouths, which are especially difficult for a Deep Decoder as they have a complicated structure contained in a small spatial extent.

We show a sample of the reconstructed images for $m = 2500$ in Figure 4, along with a breakdown of the component images from the GAN and Deep Decoder components of the hybrid model for an individual image. More samples are available in the supplemental material.

We next compare the performance of the GAN Prior, Deep Decoder, and hybrid model on images which are *outside* the training distribution of the GAN prior, shown in Figure 5. We use the same BEGAN trained on CelebA faces, but in this experiment, we reconstruct images of birds from the Caltech-USD Birds 200 dataset (Welinder et al., 2010). The hybrid model's performance is no longer significantly better than the Deep Decoder, because the GAN prior is unable to represent useful image features for this new distribution. Additionally, we observe that the coefficient $\alpha$ of the GAN prior in the mixture $H$ is diminished, so that the hybrid model "filters out" the irrelevant information from the GAN prior.

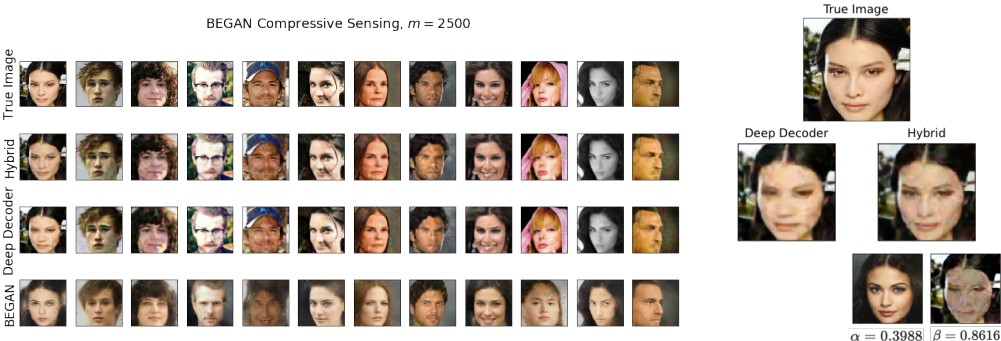

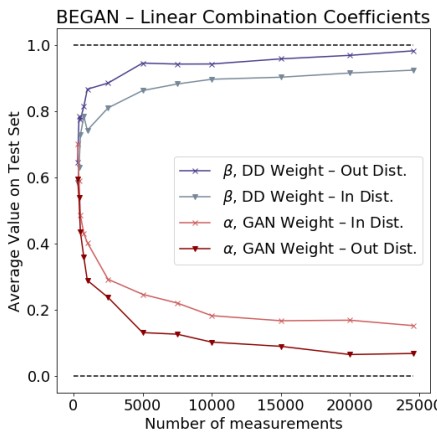

**Figure 4:** *Left*: Samples of reconstructed images for $m = 2500$ measurements, a compression ratio of $0.051$. The GAN prior has significant representation error, to the point where it appears to recover the face of the wrong person. The Deep Decoder has artifacts arising from too much smoothing of facial features. The hybrid model has sharply defined features, as does the GAN, without the unnecessary smoothness of the Deep Decoder by itself. *Right*: A comparison of output examples for the Deep Decoder and hybrid models, along with the two components underlying the hybrid model. The difference is detail is noticeable between the pure Deep Decoder and the Hybrid model.

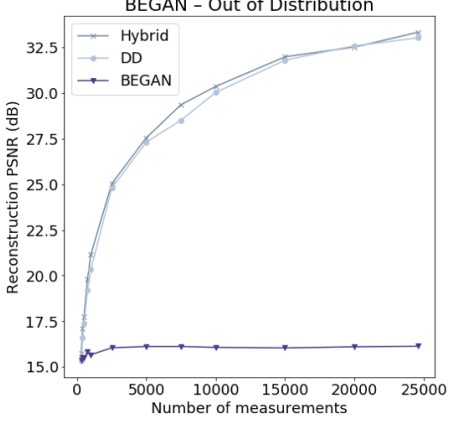

**Figure 5:** *Left*: Performance of various image models on compressed sensing of images of birds, using a GAN prior trained to generate images of celebrity faces. Surprisingly, the hybrid model still marginally outperforms the Deep Decoder, as the GAN prior learns some general image statistics which are beneficial. *Right*: Coefficients of the GAN and Deep Decoder in the hybrid $H$. Darker colors correspond to images out of the GAN prior training distribution. As one would expect, the coefficient of the GAN prior is diminished when reconstructing images for which the GAN has not learned relevant features.

### 4.1.1 COMPARISON TO OVERPARAMETRIZED IMAGE MODELS

We compare our model to an overparametrized alternative, drawing from recent work on the Deep Image Prior (DIP) (Ulyanov et al., 2018) and Image Adaptive GAN (IAGAN) (Abu Hussein et al., 2019). The Deep Image Prior is an untrained encoder-decoder architecture with fixed input whose weights are optimized from random initialization to minimize (1). The IAGAN begins with a GAN prior $G_\phi(z)$ and optimizes only over $z$ to find the latent code $z^*$ so that $G_\phi(z^*)$ minimizes (1). Then, they reduce representation error by jointly optimizing over the GAN weights $\phi$ learned from training data and the latent code, initialized at $z^*$. In contrast to the DIP, IAGAN uses only a decoder. Both architectures use orders of magnitude more parameters than the dimensionality of their output, so they are vastly overparametrized when fitting a single image. Our BEGAN has roughly $18 \times 10^6$ parameters, and we find it can fit measurements in all regimes with near perfect accuracy.

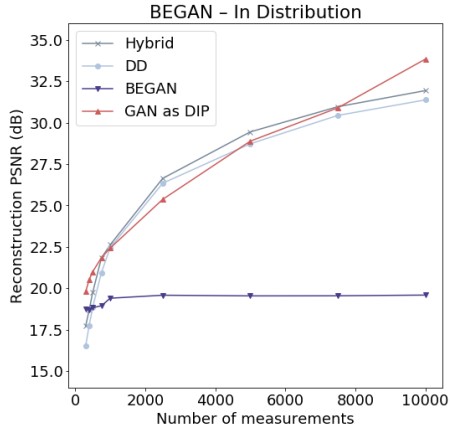 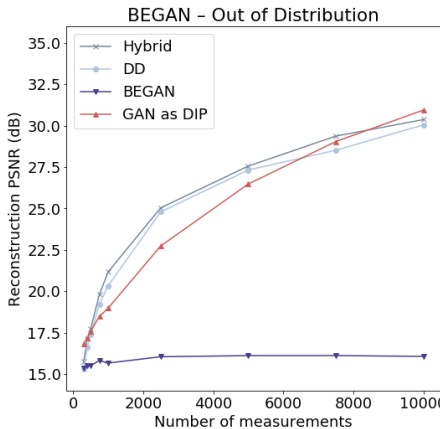

Figure 6: *Left*: Reconstruction quality for images of celebrity faces, using GAN priors trained on celebrity faces. There is a range of measurement regimes where the hybrid model outperforms the GAN as DIP model, which has significantly many more parameters. The GAN as DIP method uses significantly more parameters in its image representation, since one must optimize over all weights of a GAN. In the low measurement regime, this has no benefit in comparison to the underparametrized hybrid model. *Right*: Reconstruction quality for images of birds, using GAN priors trained on celebrity faces. In this case of out-of-distribution images, the range of measurements for which the hybrid model outperforms the GAN as DIP is is even more pronounced, demonstrating the versatility of the hybrid method.

We compare our hybrid model to an overparametrized alternative, which we call "GAN as DIP." The GAN as DIP method is similar to IAGAN in that we begin with a GAN prior and optimize a latent code $z^*$ which minimizes (1). We then optimize over the weights and latent code of the GAN prior to fit an image. However, we break from the IAGAN method in that we choose to omit any post-inversion steps, such as BM3D denoising or backprojection (Abu Hussein et al., 2019). This method is similar to DIP in that it is an overparametrized CNN, but it is not an encoder-decoder architecture. Hyperparameter details may be found in Section 6.1 of the Supplemental Materials.

We observe that that our hybrid image model outperforms the GAN as DIP for a certain range of $m$, excluding the case of extremely few measurements and the case of very many meausrements. In this range, it can outperform GAN as DIP by around 1.5 dB. This performance improvement is notable because the hybrid model has significantly fewer parameters than the GAN as DIP.

## 5 DISCUSSION

In this paper, we introduced a new hybrid model for natural images, consisting of a trained part and an untrained part; specifically our model linearly combines a GAN and a Deep Decoder. We demonstrate that this hybrid model yields higher PSNRs at compressive sensing than either the underlying GAN model (by 10+ dB) or the underlying Deep Decoder (by 1-2 dB) at all but the most extreme levels of undersampling, even when tested on images that in-distribution relative to the distribution the GAN was trained on. Interestingly, we further demonstrate that the hybrid model maintains a slight edge over the Deep Decoder even on out-of-distribution images. Similar performance is demonstrated for image superresolution. A natural point of comparison of this hybrid model is the IAGAN, which uses a trained GAN as a warm start to a Deep Image Prior. Our model can exhibit improved PSNRs by 2.5 dB over the IAGAN at appropriate undersampling ratios in compressive sensing. This work illustrates that complicated or expensive workarounds are not needed in order to build an image model that is as good as trained generators and as good as untrained models. It is unexpected that simply taking the linear combination of these two models would yield benefits over both models separately.

The strength of the proposed hybrid model derives from how the strengths of the unlearned deep decoder compensate for the weaknesses of the GAN and vice versa. The primary weakness of GANs is an inherent representation error, particular significant for (slight) out of distribution images, that

can be attributed primarily to the fact that the prior pertains to a particular distribution of images. The primary strength of a Deep Decoder is that it has minimal representation error for natural images, and no bias towards on natural images over the other. The Deep Decoder is an underparameterized neural network that has been empirically shown to be effective at modeling natural images while incapable of fitting noise (unlike the Deep Image Prior, which consequently requires early stopping). However, since the deep decoder does not incorporate any learned information of a particular image class, it can't exploit such additional information. Our hybrid model appears to combine the best from the two worlds: it has extremely small representation error, inherited from the deep decoder component, but at the same time enables exploiting potential prior knowledge about an image class. As the deep decoder is unlearned, it is not tied to any particular training distribution, which is a strength, but it consequently loses out on improvements that should be possible from training. For example, a deep decoder may smooth out regions of an image where there should be edges, as the Deep Decoder does not know from learning that edges may be natural. These improvements are exactly what the GAN provides in the hybrid model.

Philosophically, the hybrid model views the GAN as providing a base image and views the deep decoder as learning a residual between that base image and what is needed to fit provided measurements. As we see, often the GAN outputs a face that is similar to the target face, but with incorrect skin hues, hair details, etc., which the Deep Decoder then corrects. Empirically, we see cases where the output of the GAN component of the hybrid model looks like the a different person, but the hybrid model looks like the same person as the target image.

| *Parameter Counts* | Hybrid | DD | GAN | GAN as Deep Image Prior |
|---|---|---|---|---|
| BEGAN | 19583 | 19519 | 64 | 2592192 |
| DCGAN | 9644 | 9544 | 100 | 3576704 |

Figure 7: Parameter counts of image representations for image models used in our compressed sensing experiments. We report the *maximum* parameter count of the Deep Decoder and hybrid models throughout our experiments, but the actual parameter count varies so that the Deep Decoder and hybrid models remain underparametrized with respect to the number of measurements. The BEGAN models represent 128px images, with 49152 degrees of freedom. The DCGAN model represents 64px images, with 12288 degrees of freedom.

One strength of our proposed hybrid model is that it has relatively few parameters. This makes computations particular cheap and would be expected to improve robustness to noise, as was noticed with the plain Deep Decoder. In the experiments we report, the hybrid model is underparameterized in that it has fewer parameters to optimized (once its GAN component is fixed) than the number of pixels in the image. See Figure 7 for the number of parameters for each of the approaches we studied. The balance between having very small representation error while being underparameterized primarily stems from the Deep Decoder component of the hybrid model. Now we can compare to alternative approaches to having small or zero representation error. One approach for inverse problems which has zero representation error as been using trained invertible neural networks as image priors. Unfortunately, because such networks are fully invertible, they have an extreme number of parameters to optimize, resulting in very expensive optimization problems. Another approach is to use the trained generative model as a warm start to a Deep Image prior (the IAGAN). While cheaper than invertible networks, this model still is significantly overparameterized because all of the generator weights can be updated at inversion time. In comparison to both of these perspectives, our proposed hybrid model has significantly fewer parameters to optimize.

This work gets at an important question in the field of inverse problems using generative models: how can we capitalize on the benefits of learning a signal distribution with the benefits of unlearned methods that apply to many signal distributions. The methods we use for inversion in practice should be at least as good as both of these categories of methods across a variety of use cases (low number of measurements vs. high number of measurements, low noise vs. high noise, variations in amount of training data available). This paper shows that it is possible to combine the benefits of both learned and unlearned methods when solving problems both in-distribution and out-of-distribution.

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

# 6 APPENDIX

## 6.1 HYPERPARAMETERS

We minimize (1) iteratively using the Adam optimizer (Kingma & Ba, 2014). Unless otherwise stated, all experiments use a learning rate $\alpha = 10^{-2}$, $\beta_1 = 0.9$, and $\beta_2 = 0.999$.

To invert the GAN as DIP model, we use the same conditions as the hybrid model, using $n_{\text{pre}} = 500$ iterations to find $z^*$ and then $n = 5000$ iterations optimizing over the latent code and all weights of the GAN prior. We use a significantly smaller learning rate, $\alpha = 10^{-4}$, to optimize the weights of the GAN prior.

## 6.2 SAMPLE SHEETS

BEGAN Compressive Sensing, $m = 24576$

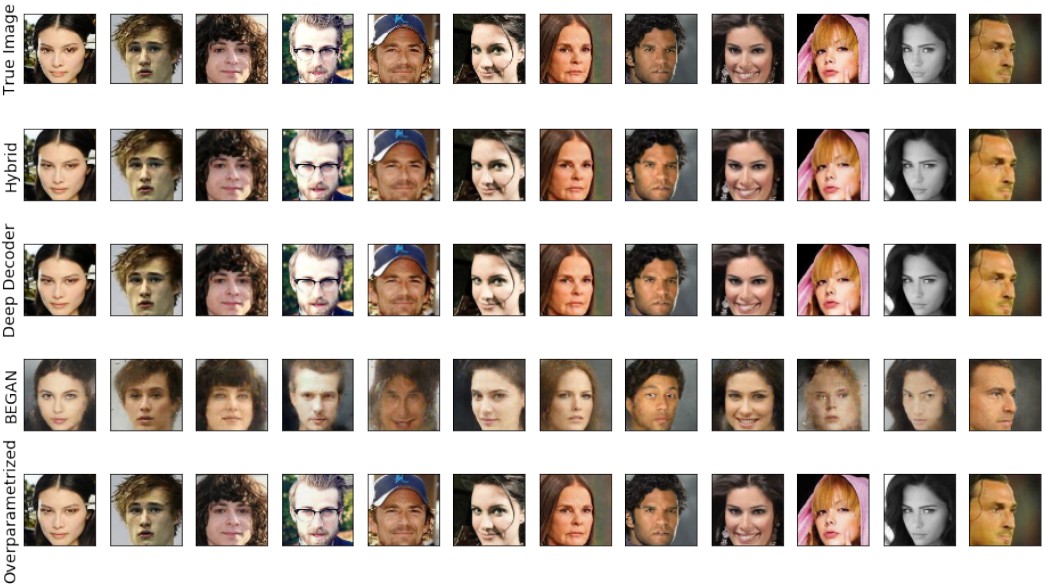

BEGAN Compressive Sensing, $m = 7500$

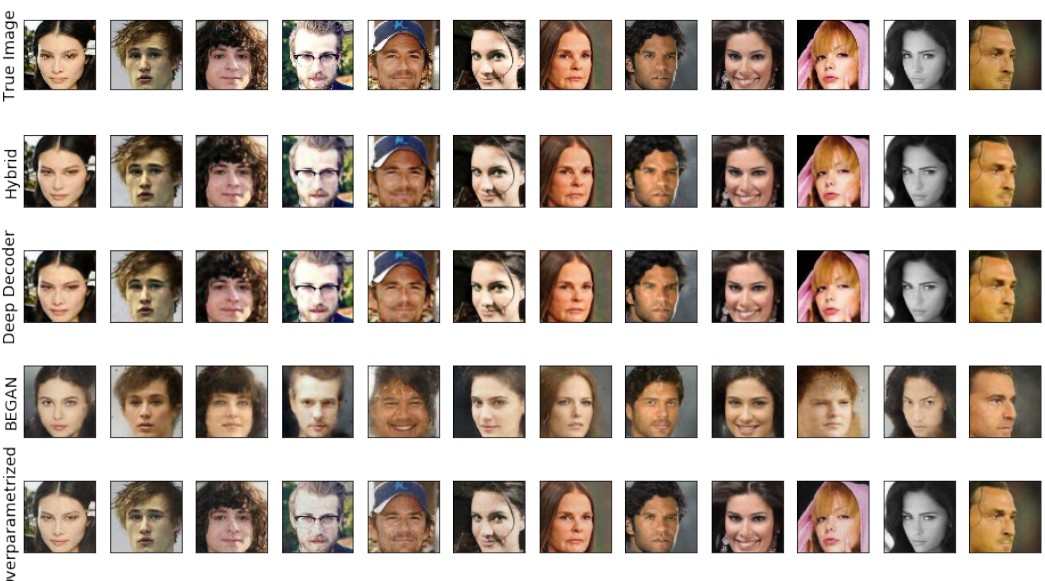

BEGAN Compressive Sensing, $m = 5000$

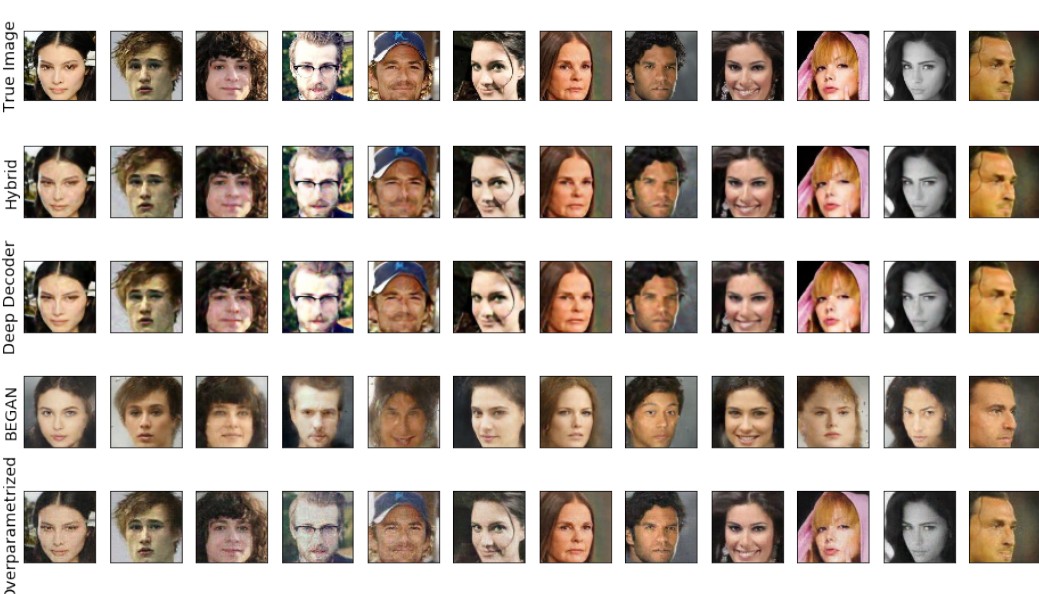

BEGAN Compressive Sensing, $m = 1000$

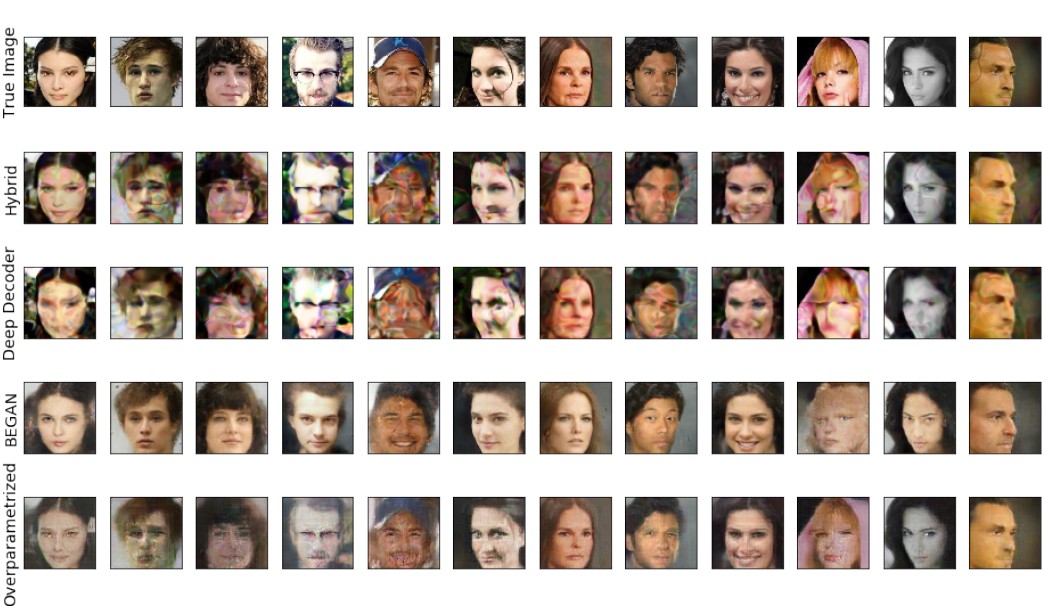

