# OpenReview forum: "Removing the Representation Error of GAN Image Priors Using the Deep Decoder"
_ICLR.cc/2020/Conference — Reject_

### Official Review · AnonReviewer1 · 2019-10-23
**Official Blind Review #1**

**Rating:** 1

**Review:**

[Update after rebuttal period]
I am sorry that the response cannot address my confusion. I still doubt the motivation of this paper and the actual experimental performance compared with state-of-the-art methods are still ignored. Thus I decrease my score.

[Original reviews]
This paper proposed to modeling image as the combination of a GAN with a Deep Decoder, to remove the representation error of a GAN when used as a prior in inverse problems. The proposed methods are evaluated on two image restoration tasks, including compressive sensing and image super-resolution. The effectiveness of the combination is also presented.

Authors devote themselves to remove the representation error of the GAN image prior. Intuitively, the manner of the proposed linear combination model is rough and less reasonable. In Alg.1, the detailed algorithmic process is presented, it is clear that authors need to pre-train the used GAN and Deep Decoder, then combine them to train one network. If the motivation of this paper is to remove the representation error of GAN, GAN should be viewed as the main body. However, the authors view GAN and Deep Decoder as the same position against the original intention.

Additionally, in the experimental part, the ablation studies indeed reflect the effectiveness of the proposed algorithm, but it looks like the Deep Decoder plays a key role in all cases. In other words, the GAN image prior just plays a supporting role. This is also far away from motivation.

More importantly, I cannot see the surprising results because of this work only compare themselves with some basic version or naïve methods. All state-of-the-art approaches to different tasks are ignored, which is the other big disadvantage.

In Table 2, what is the meaning of ‘CSGM’？ The authors should describe it.

In a word, from the algorithmic and experimental perspective, this paper cannot achieve satisfying performance.


**Experience Assessment:**

I have published in this field for several years.

**Review Assessment: Checking Correctness Of Derivations And Theory:**

N/A

**Review Assessment: Checking Correctness Of Experiments:**

I carefully checked the experiments.

**Review Assessment: Thoroughness In Paper Reading:**

I read the paper at least twice and used my best judgement in assessing the paper.

---

> ### Author Response · Authors · 2019-11-14
> **Response to Reviewer 1**
>
> Thank you for your comments and corrections. We have updated table 2 to use "GAN" instead of "CSGM" to reflect that the value for that column represents the dimensionality of each GAN Prior's latent representation.
>
> In response to your comments about the motivation of our model: our goal in this paper is to propose a simple model which can leverage benefits from a GAN Prior's learned approximation of the data distribution, without being tied to that distribution. The dependence of a GAN prior on its data distribution is a significant flaw, where we would ideally like to have strictly improved performance given examples of the class of images being recovered.
>
> So, it is intentional that the GAN only plays a supporting role, for those images where is has learned useful representations.
>
> We demonstrate that our model significantly reduces the representation error of the GAN, and comparisons to the Deep Decoder only indicate that the GAN Prior contributes its own benefits where it is possible. On out of distribution images, it is expected that the GAN Prior would be useless – our model is robust to this failure.
>
> We appreciate your comments and look forward to improving this method further.

---

### Official Review · AnonReviewer3 · 2019-10-25
**Official Blind Review #3**

**Rating:** 3

**Review:**

Summary: This paper proposes to use a combination of a pretrained GAN and an untrained deep decoder as the image prior for image restoration problem. The combined model jointly infers the latent code for the trained GAN and the parameters in the untrained deep decoder. It also jointly infers the mixing coefficient alpha and beta during test time for each image, thus learning how much we should rely on GAN. The proposed hybrid model is helpful on compressed sensing experiments on the CelebA dataset; however, it is only marginally better than deep decoder on image super resolution and out-of-distribution compressed sensing.

Detailed comments:
-	The writing is clear and I was able to understand the model part of the paper. The algorithm box is helpful. However, I would still appreciate if the authors can provide an overall model figure in the model section to help understanding.
-	Jointly learning the mixing coefficient is an interesting part of the model.
-	The motivation in the abstract and intro could be strengthened. A smaller version of Figure 1 can be probably moved to the beginning of the paper to illustrate the problem of GAN. But even with the help of Figure 1, it is still unclear what is the fundamental problem for GAN. Simply combining a GAN with an untrained decoder model doesn’t help elucidate the source of the problem.
-	The proposed Hybrid model seems to help on compressed sensing experiments on CelebA. However, it doesn’t help much on out-of-distribution experiments. Moreover, in the super-resolution task, as shown in Figure 5, the improvement over deep decoder is also not significant.
-	The out-of-distribution experiments seems lack of thorough study. In particular, the paper only studies the transfer between CelebA -> Caltech-UCSD Bird dataset. It would be better if the paper can study a variety of other image datasets as well. Also some visualization on the Bird dataset should also be included.
-	Effect of n_pre needs to be further investigated. Why not directly train both models together? It would be good if the authors could comment on how sensitive the n_pre is and what is the intuition.
-	For figures, I would recommend rename “Hybrid” to “Hybrid (Ours)” to highlight the paper’s contribution, and use a brighter color.
-	Figure 5 should be renamed as a Table.
-	Hyperparameter details should be moved to the Experiment section.

Conclusion:
The paper proposes a simple combination of a trained GAN and an untrained decoder model for the task of image restoration. Although the method is clear and straightforward, in the experiments, the influence of the new model component seems marginal. Moreover, the motivation is not strong enough. Therefore, I recommend weak reject.

**Experience Assessment:**

I have read many papers in this area.

**Review Assessment: Checking Correctness Of Derivations And Theory:**

N/A

**Review Assessment: Checking Correctness Of Experiments:**

I assessed the sensibility of the experiments.

**Review Assessment: Thoroughness In Paper Reading:**

I read the paper at least twice and used my best judgement in assessing the paper.

---

> ### Author Response · Authors · 2019-11-14
> **Response to Reviewer 3**
>
> Thank you for your comments and corrections. We have updated the paper with a diagram of our model, and moved the hyperparameter descriptions to a section of the appendix.
>
> Our intention in the superresolution section is to demonstrate that our hybrid model is applicable for inverse problems besides compressed sensing. However, we agree with your concerns that improvement of the hybrid model over the deep decoder is not very significant. Since we already demonstrate that the hybrid model is no worse than a deep decoder via the out of distribution experiments, we have decided to remove the superresolution section from the paper.
>
> In response to your concerns about n_pre, we observed that when n_pre is set to zero, there is some variance in the experimental results. The intuition is that both the GAN and Deep Decoder are randomly initialized, and so they can interfere with each other early in the inversion procedure. The results are otherwise not sensitive to choice of n_pre (within the same order of magnitude)

---

### Official Review · AnonReviewer2 · 2019-10-26
**Official Blind Review #2**

**Rating:** 3

**Review:**

This paper presents a method for reducing the representation error generative convolutional neural networks by combining them with untrained deep decoder. The method is evaluated on compressive sensing and super-resolution, where a better performance than the isolated use of Deep Decoders and GAN priors. The main contribution of the paper is not the performance, but the simplicity of this approach.



For the title, I would suggest to replace the word Removing with Reducing.
Furthermore, the clarification of "GAN prior" is very nice in the introduction, maybe you could already clarify it in the abstract.

You should perform a critical grammar check. There are too many commas, for example:
"At sufficiently difficult superresolution problems, the Hybrid model outperforms, the Deep
Decoder, Bicubic upsampling, the BEGAN prior, and the BEGAN as DIP prior." -> there should be no comma after "outperforms"
The sentence from Page 3 to 4 reads strangely, probably a word is missing after "For our GAN prior, we use the BEGAN architecture, and we demonstrate similar results"
"Philosophically, they hybrid" -> "Philosophically, the hybrid"

Fig 6 caption - shouldn't it be 49152 instead of 49512?

You perform various very good analysis experiments, which is well appreciated. Still, it would be good to think about some more experiments (and include at least one of them in the paper):
1. You compare to IGAN and show that you achieve similar performance. You describe that a state-of-the-art approach are invertible generative models and that they are very time consuming (e.g., 15 minutes for a 64x64 image). How good would the invertible models be in terms of performance? Could you perform tests as well?
2. It would be great if you report the runtime of all experiments as well - maybe also the memory usage.

**Experience Assessment:**

I have published one or two papers in this area.

**Review Assessment: Checking Correctness Of Derivations And Theory:**

I assessed the sensibility of the derivations and theory.

**Review Assessment: Checking Correctness Of Experiments:**

I carefully checked the experiments.

**Review Assessment: Thoroughness In Paper Reading:**

I read the paper at least twice and used my best judgement in assessing the paper.

---

> ### Author Response · Authors · 2019-11-14
> **Response to Reviewer 2**
>
> Thank you for your comments and corrections. We have fixed the typos you pointed out.
>
> In response to your questions:
> 1. Anecdotally, we observe the invertible models perform significantly better (4-6 dB psnr) in most measurement regimes (> 2% measurement ratio). The improvement is slightly less for lower measurement ratios (1-2 db psnr). This comes at a cost of significantly more expensive inversion, and a larger representation size.
> 2. In our experiments, the GAN as DIP, Deep Decoder, and Hybrid models all take roughly 200 seconds (3.33 minutes).

---

### Decision · Program_Chairs · 2019-12-19

**Decision:**

Reject

**Comment:**

The paper introduces a method for removing what they call representation error and apply the method to super resolution and compressive sensing.

The reviewers have provided constructive feedback. The reviewers like aspects of the paper but are also concerned with various shortcomings. The consensus is that the paper is not ready for publication as it stands.

Rejection is therefore recommended with strong encouragement to keep working on the method and submit elsewhere.